# High-Fidelity Generative Image Compression

**Fabian Mentzer**[*]  **George Toderici**  **Michael Tschannen**[†]  **Eirikur Agustsson**
ETH Zürich      Google Research      Google Research      Google Research

## Abstract

We extensively study how to combine Generative Adversarial Networks and learned compression to obtain a state-of-the-art generative lossy compression system. In particular, we investigate normalization layers, generator and discriminator architectures, training strategies, as well as perceptual losses. In contrast to previous work, i) we obtain visually pleasing reconstructions that are perceptually similar to the input, ii) we operate in a broad range of bitrates, and iii) our approach can be applied to high-resolution images. We bridge the gap between rate-distortion-perception theory and practice by evaluating our approach both quantitatively with various perceptual metrics, and with a user study. The study shows that our method is preferred to previous approaches even if they use more than $2\times$ the bitrate.

## 1  Introduction

The ever increasing availability of cameras produces an endless stream of images. To store them efficiently, lossy image compression algorithms are used in many applications. Instead of storing the raw RGB data, a lossy version of the image is stored, with—hopefully—minimal visual changes to the original. Various algorithms have been proposed over the years [50, 43, 54], including using state-of-the-art video compression algorithms for single image compression (BPG [7]). At the same time, deep learning-based lossy compression has seen great interest [45, 5, 31], where a neural network is directly optimized for the rate-distortion trade-off, which led to new state-of-the-art methods.

However, all of these approaches degrade images significantly as the compression factor increases. While classical algorithms start to exhibit algorithm-specific artifacts such as blocking or banding, learning-based approaches reveal issues with the distortion metric that was used to train the networks. Despite the development of a large variety of "perceptual metrics" (*e.g.*, (Multi-Scale) Structural Similarity Index ((MS-)SSIM) [53, 52], Learned Perceptual Image Patch Similarity (LPIPS) [57]), the weakness of each metric is exploited by the learning algorithm, *e.g.*, checkerboard artifacts may appear when targeting neural network derived metrics, relying on MS-SSIM can cause poor text reconstructions, and MSE yields blurry reconstructions.

In [3], Agustsson *et al.* demonstrated the potential of using GANs to prevent compression artifacts with a compression model that produces perceptually convincing reconstructions for extremely low bitrates (<0.08 bpp). However, their reconstructions tend to only preserve high-level semantics, deviating significantly from the input.

Recent work by Blau and Michaeli [9] characterized this phenomenon by showing the existence of a triple "rate-distortion-perception" trade-off, formalizing "distortion" as a similarity metric comparing pairs of images, and "perceptual quality" as the distance between the image distribution $p_X$ and the distribution of the reconstructions $p_{\hat{X}}$ produced by the decoder, measured as a distribution divergence. They show that at a fixed rate, better perceptual quality always implies worse distortion. Conversely, only minimizing distortion will yield poor perceptual quality. To overcome this issue, distortion can

---

[*]Work done while interning at Google Research.      Project page and demo: `hific.github.io`
[†]Work done while at Google Research.      Correspondence: eirikur@google.com

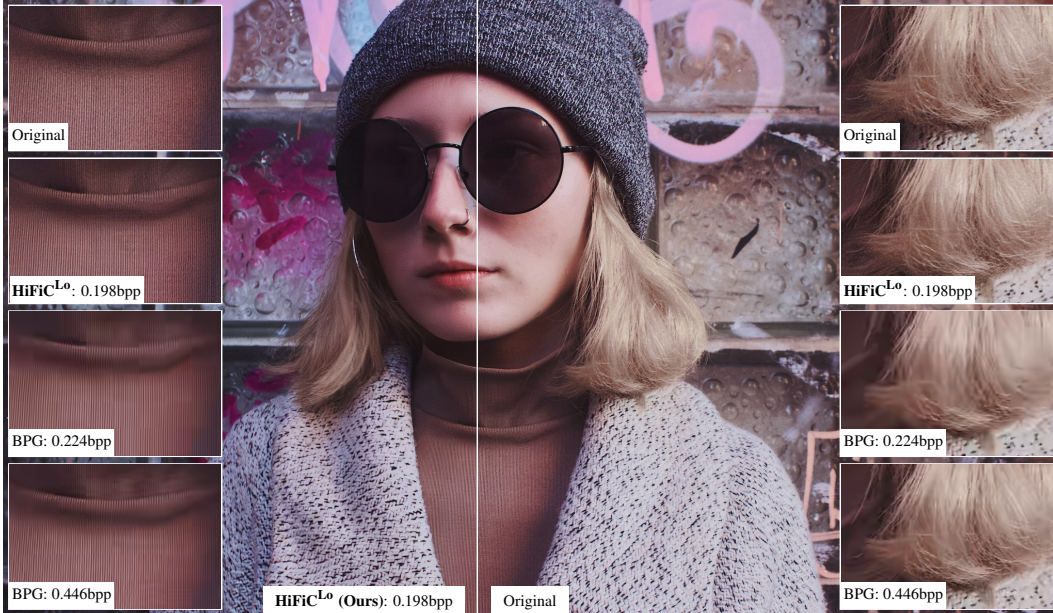

Figure 1: Comparing our method, *HiFiC*, to the original, as well as BPG at a similar bitrate and at $2\times$ the bitrate. We can see that our GAN model produces a high-fidelity reconstruction that is very close to the input, while BPG exhibits blocking artifacts, that are still present at double the bitrate. In the background, we show a split to the original to further indicate how close our reconstruction is. We show many more visual comparisons in Appendix B, including more methods, more bitrates, and various datasets. *Best viewed on screen.*

be traded for better perceptual quality by minimizing the mismatch between the distributions of the input and the reconstruction, *e.g.*, with Generative Adversarial Networks (GANs) [17]. While [9] presents comprehensive theoretical results, the rate-distortion-perception trade-off is only explored empirically on toy datasets.

In this paper, we address these issues with the following contributions:

1. We propose a generative compression method to achieve high quality reconstructions that are very close to the input, for high-resolution images (we test up to $2000{\times}2000$ px). In a user study, we show that our approach is visually preferred to previous approaches *even when these approaches use more than $2\times$ higher bitrates*, see Fig. 3.

2. We quantitatively evaluate our approach with FID [18], KID [8], NIQE [35], LPIPS [57], and the classical distortion metrics PSNR, MS-SSIM, and show how our results are consistent with the rate-distortion-perception theory. We also show that no metric would have predicted the full ranking of the user study, but that FID and KID are useful in guiding exploration. Considering this ensemble of diverse perceptual metrics including no-reference metrics, pair-wise similarities, and distributional similarities, as well as deep feature-based metrics derived from different network architectures, ensures a robust perceptual evaluation.

3. We extensively study the proposed architecture and its components, including normalization layers, generator and discriminator architectures, training strategies, as well as the loss, in terms of perceptual metrics and stability.

## 2 Related work

The most frequently used lossy compression algorithm is JPEG [50]. Various hand-crafted algorithms have been proposed to replace JPEG, including WebP [54] and JPEG2000 [43]. Relying on the video codec HEVC [42], BPG [7] achieves very high PSNR across varying bitrates. Neural compression approaches directly optimize Shannon's rate-distortion trade-off [14]. Initial works relied on RNNs [45, 47], while subsequent works were based on auto-encoders [5, 44, 1]. To decrease the required bitrate, various approaches have been used to more accurately model the probability

density of the auto-encoder latents for more efficient arithmetic coding, using hierarchical priors, auto-regression with various context shapes, or a combination thereof [6, 31, 28, 39, 32, 26, 33]. State-of-the-art models now outperform BPG in PSNR, *e.g.*, the work by Minnen *et al.* [32].

Since their introduction by Goodfellow *et al.* [17], GANs have led to rapid progress in unconditional and conditional image generation. State-of-the-art GANs can now generate photo-realistic images at high resolution [10, 21, 38]. Important drivers for this progress were increased scale of training data and model size [10], innovation in network architectures [21], and new normalization techniques to stabilize training [36]. Beyond (un)conditional generation, adversarial losses led to advances in different image enhancement tasks, such as compression artifact removal [15], image de-noising [11], and image super-resolution [25]. Furthermore, adversarial losses were previously employed to improve the visual quality of neural compression systems [39, 40, 48, 3, 9]. [39] uses an adversarial loss as a component in their full-resolution compression system, but they do not systematically ablate and asses the benefits of this loss on the quality of their reconstructions. While [40] provides a proof-of-concept implementation of a low-resolution compression system with a GAN discriminator as decoder, [48, 9] focus on incorporating a GAN loss into the rate-distortion objective in a conceptually sound fashion. Specifically, [48] proposes to augment the rate-distortion objective with a distribution constraint to ensure that the distribution of the reconstructions match the input distribution at all rates, and [9] introduce and study a triple trade-off between rate, distortion, and distribution matching. Finally, [3] demonstrates that using GAN-based compression systems at very low bitrates can lead to bitrate savings of $2\times$ over state-of-the-art engineered and learned compression algorithms.

## 3 Method

### 3.1 Background

**Conditional GANs** Conditional GANs [17, 34] are a way to learn a generative model of a conditional distribution $p_{X|S}$, where each datapoint $x$ is associated with additional information $s$ (*e.g.*, class labels or a semantic map) and $x, s$ are related through an unknown joint distribution $p_{X,S}$. Similar to standard GANs, in this setting we train two rivaling networks: a generator $G$ that is conditioned on $s$ to map samples $y$ from a fixed known distribution $p_Y$ to $p_{X|S}$, and a discriminator $D$ that maps an input $(x, s)$ to the probability that it is a sample from $p_{X|S}$ rather than from the output of $G$. The goal is for $G$ to "fool" $D$ into believing its samples are real, *i.e.*, from $p_{X|S}$. Fixing $s$, we can optimize the "non-saturating" loss [17]:

$$\mathcal{L}_G = \mathbb{E}_{y\sim p_Y}[-\log(D(G(y,s),s)],$$
$$\mathcal{L}_D = \mathbb{E}_{y\sim p_Y}[-\log(1 - D(G(y,s),s)] + \mathbb{E}_{x\sim p_{X|s}}[-\log(D(x,s))]. \tag{1}$$

**Neural Image Compression** Learned lossy compression is based on Shannon's rate-distortion theory [14]. The problem is usually modeled with an auto-encoder consisting of an encoder $E$ and a decoder $G$. To encode an image $x$, we obtain a quantized latent $y = E(x)$. To decode, we use $G$ to obtain the lossy reconstruction $x' = G(y)$. The compression incurs a distortion $d(x, x')$, *e.g.*, $d = \text{MSE}$ (mean squared error). Storing $y$ is achieved by introducing a probability model $P$ of $y$. Using $P$ and an entropy coding algorithm (*e.g.*, arithmetic coding [30]), we can store $y$ losslessly using bitrate $r(y) = -\log(P(y))$ (in practice there is a negligible bit-overhead incurred by the entropy coder). If we parameterize $E, G$, and $P$ as CNNs, we can train them jointly by minimizing the *rate-distortion trade-off*, where $\lambda$ controls the trade-off:

$$\mathcal{L}_{EG} = \mathbb{E}_{x\sim p_X}[\lambda\, r(y) + d(x,x')]. \tag{2}$$

### 3.2 Formulation and Optimization

We augment the neural image compression formulation with a conditional GAN, *i.e.*, we merge Eqs. 1, 2 and learn networks $E, G, P, D$. We use $y = E(x)$, and $s = y$. Additionally, we use the "perceptual distortion" $d_P = \text{LPIPS}$, inspired by [51], who showed that using a VGG [41]-based loss helps training. In the formalism of [9], $d_P$ is a distortion (as it is applied point-wise), thus we group it together with MSE to form our distortion $d = k_M\text{MSE} + k_P d_P$, where $k_M, k_P$ are hyper-parameters. Using hyper-parameters $\lambda, \beta$ to control the trade-off between the terms, we obtain:

$$\mathcal{L}_{EGP} = \mathbb{E}_{x\sim p_X}[\lambda r(y) + d(x,x') - \beta\log(D(x',y))], \tag{3}$$
$$\mathcal{L}_D = \mathbb{E}_{x\sim p_X}[-\log(1 - D(x',y))] + \mathbb{E}_{x\sim p_X}[-\log(D(x,y))]. \tag{4}$$

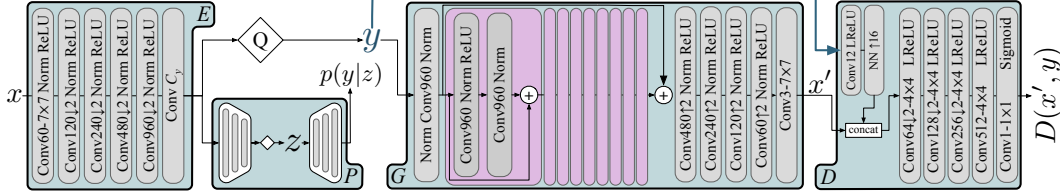

Figure 2: Our architecture. *ConvC* is a convolution with $C$ channels, with $3{\times}3$ filters, except when denoted otherwise. $\downarrow 2$, $\uparrow 2$ indicate strided down or up convolutions. *Norm* is ChannelNorm (see text), *LReLU* the leaky ReLU [56] with $\alpha{=}0.2$, *NN*↑16 nearest neighbor upsampling, $Q$ quantization.

**Constrained Rate**   When training a neural compression model w.r.t. the loss in Eq. 2, there is only a single term, $d(x, x')$, that is at odds with the rate term $r(y)$. The final (average) bitrate of the model can thus be controlled by varying only $\lambda$. In our setting however, MSE, $d_P$, and $-\log(D(x'))$ are at odds with the rate. For a fixed $\lambda$, different $k_M, k_P, \beta$ would thus result in models with different bitrates, making comparison hard. To alleviate this, we introduce a "rate target" hyper-parameter $r_t$, and replace $\lambda$ in Eq. 3 with an *adaptive* term $\lambda'$ that uses the two hyper parameters $\lambda^{(a)}, \lambda^{(b)}$, where $\lambda' = \lambda^{(a)}$ if $r(y) > r_t$, and $\lambda' = \lambda^{(b)}$ otherwise. Setting $\lambda^{(a)} \gg \lambda^{(b)}$ allows us to learn a model with an average bitrate close to $r_t$.

### 3.3   Architecture

We show our architecture in Fig. 2, including the encoder $E$, generator $G$, discriminator $D$ and probability model $P$. For $P$, we use the hyper-prior model proposed in [6], where we extract side information $z$ to model the distribution of $y$ and simulate quantization with uniform noise $\mathcal{U}(-1/2, 1/2)$ in the hyper-encoder and when estimating $p(y|z)$. However, when feeding $y$ to $G$ we use rounding instead of noise (*i.e.*, the straight-through estimator as in [44]), which ensures that $G$ sees the same quantization noise during training and inference. $E, G$ and $D$ are based on [51, 3], with some key differences in the discriminator and in the normalization layers, described next.

Both [51, 3] use a multi-scale patch-discriminator $D$, while we only use a single scale, and we replace InstaceNorm [49] with SpectralNorm [36]. Importantly, and in contrast to [3], we condition $D$ on $y$ by concatenating an upscaled version to the image, as shown in Fig. 2. This is motivated from using a conditional GAN formulation, where $D$ has access to the the conditioning information (which for us is $y$, see Section 3.2).

In [51] InstanceNorm is also used in $E, G$. In our experiments, we found that this introduces significant darkening artifacts when employing the model on images with a different resolution than the training crop size (see Section 5.3). We assume this is due to the spatial averaging of InstanceNorm, and note that [38] also saw issues with InstanceNorm. To alleviate these issues, we introduce **ChannelNorm**, which normalizes over channels. The input is a batch of $C{\times}H{\times}W$-dimensional feature maps $f_{chw}$, which are normalized to

$$f'_{chw} = \frac{f_{chw} - \mu_{hw}}{\sigma_{hw}} \alpha_c + \beta_c, \quad \text{where} \quad \mu_{hw} = {}^1\!/C \sum_{c=1}^{C} f_{chw} \tag{5}$$

$$\sigma_{hw}^2 = {}^1\!/C \sum_{c=1}^{C} (f_{chw} - \mu_{hw})^2,$$

using learned per-channel offsets $\alpha_c, \beta_c$. We note that ChannelNorm has similarities to other normalizations. Most closely related, Positional Normalization [27] also normalizes over channels, but does not use learned $\alpha_c, \beta_c$. Instead, they propose to use $\mu, \sigma$ from earlier layers as offsets for later layers, motivated by the idea that these $\mu, \sigma$ may contain useful information. Further, Kerras *et al.* [20] normalize each feature vector "pixelwise" to unit length, without centering ($\mu = 0$). We show a visualization and more details in Appendix A.4.

### 3.4   User Study

To visually validate our results, we set up a user study as two-alternative forced choice (2AFC) on $N_I{=}20$ random images of the CLIC2020 [46] dataset (datasets are described below), inpsired by the methodology used in [46]. Participants see crops of these images, which are determined as follows: when an image is first shown, a random $768{\times}768$ crop is selected. The participants are asked to request a new random crop until they find "an interesting crop" (*i.e.*, a region that is not completely

flat or featureless), and then they use this crop for all comparisons on that image. We chose the crop size such that it is as large as possible while still allwoing two crops to fit side-by-side on all screens, downsampling would cause unwanted biases. The crops selected by users are linked in Appendix A.5.

At any time, the participants are comparing a pair of methods A and B. On screen, they always see two crops, the left is either method A or B, the right is always the original (Fig. A9 shows a screenshot). Participants are asked to toggle between methods to "select the method that looks closer to the original". This ensures the participants take into account the faithfulness of the reconstruction to the original image. We select the pairs of methods to compare by performing a binary search against the ranking of previously rated images, which reduces the number of comparisons per participant.

To obtain a ranking of methods, we use the Elo [16] rating system, widely used to rank chess players. We view each pair-wise comparison as a game between A and B, and all comparisons define a tournament. Running Elo in this tournament (using Elo parameters $k = 30$ and $N = 400$), yields a score per method. Since Elo is sensitive to game order, we run a Monte Carlo simulation, shuffling the order of comparisons $N_E$=10 000 times. We report the median score over the $N_E$ simulations for each method, giving us an accurate estimate of the skill rating of each method and confidence intervals. We validated that the ordering is consistent with other methods for obtaining total orderings from pairwise comparisons that do not allow the computation of confidence intervals [37, 12].

## 4 Experiments

**Datasets**   Our training set consists of a large set of high-resolution images collected from the Internet, which we downscale to a random size between 500 and 1000 pixels, and then randomly crop to $256{\times}256$. We evaluate on three diverse benchmark datasets collected independently of our training set to demonstrate that our method generalizes beyond the training distribution: the widely used *Kodak* [23] dataset (24 images), as well as the *CLIC2020* [46] testset (428 images), and *DIV2K* [2] validation set (100 images). We note that we ignore color profiles of CLIC images. The latter two mostly contain high-resolution images with shorter dimension greater than 1300px (see Appendix A.9 for more statistics). We emphasize that we do not adapt our models in any way to evaluate on these datasets, and that we evaluate on the full resolution images.

**Metrics**   We evaluate our models and the baselines in PSNR as well as the perceptual distortions LPIPS [57] and MS-SSIM, and we use NIQE [35], FID [18], and KID [8] as perceptual quality indices. MS-SSIM is the most widely used perceptual metric to asses (and train) neural compression systems. LPIPS measures the distance in the feature space of a deep neural network originally trained for image classification, but adapted for predicting the similarity of distorted patches, which is validated to predict human scores for these distortions [57]. Three network architectures were adapted in [57], we use the variant based on AlexNet. NIQE is a no-reference metric, also based on assessing how strongly the statistics of a distorted image deviate from the statistics of unmodified natural images. In contrast to FID, non-learned features are used, and the metric entails focusing on patches selected with a "local sharpness measure".

FID is a widely used metrics to asses sample quality and diversity in the context of image generation, in particular for GANs. Unlike PSNR, MS-SSIM, and LPIPS, which measure the similarity between individual *pairs* of images, FID assesses the similarity between the *distribution* of the reference images and the distribution of the generated/distorted images. This similarity is measured in the feature space of an Inception network trained for image classification, by fitting Gaussians to the features and measuring a Wasserstein distance betweeen the Gaussians of the references and the Gaussians of the generated images. Heusel *et al.* [18] demonstrate that FID is consistent with increasing distortion and human perception. Furthermore, it was shown to detect common GAN failure modes such as mode dropping and mode collapse [29]. KID is similar, but unlike FID, is unbiased and does not make any assumptions about the distributions in the features space.

As the distortion loss ensures global consistency of the image, and due to the large variation of resolutions in our test sets, we calculate FID and KID on patches rather than on the full images, covering the image (details in Appendix A.7). We use a patch size of 256 pixels, which yields 28 650 patches for CLIC2020 and 6 573 patches for DIV2K. We do not report FID or KID for Kodak, as the 24 images only yield 192 patches.

**Model Variants and Baselines**   We call our main model *High Fidelity Compression (HiFiC)*. To see the effect of the GAN, we train a baseline with the same architecture and same distortion loss

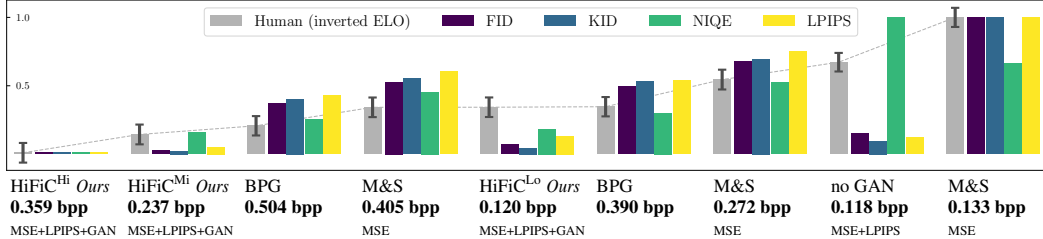

| HiFiC^Hi *Ours* | HiFiC^Mi *Ours* | BPG | M&S | HiFiC^Lo *Ours* | BPG | M&S | no GAN | M&S |
| --- | --- | --- | --- | --- | --- | --- | --- | --- |
| **0.359 bpp** | **0.237 bpp** | **0.504 bpp** | **0.405 bpp** | **0.120 bpp** | **0.390 bpp** | **0.272 bpp** | **0.118 bpp** | **0.133 bpp** |
| MSE+LPIPS+GAN | MSE+LPIPS+GAN | | MSE | MSE+LPIPS+GAN | | MSE | MSE+LPIPS | MSE |

Figure 3: Normalized scores for the user study, compared to perceptual metrics. We invert human scores such that **lower is better** for all. Below each method, we show *average* bpp, and for learned methods we show the loss components. "no GAN" is our baseline, using the same architecture and distortion $d$ as *HiFiC (Ours)*, but no GAN. "M&S" is the *Mean & Scale Hyperprior* MSE-optimized baseline. The study shows that training with a GAN yields reconstructions that outperform BPG at practical bitrates, for high-resolution images. Our model at 0.237bpp is preferred to BPG even if BPG uses $2.1\times$ the bitrate, and to MSE optimized models even if they use $1.7\times$ the bitrate.

$d = k_M\text{MSE} + k_P d_P$, but no GAN loss, called ***Baseline (no GAN)***. We train all models with Adam [22] for $2\,000\,000$ steps, and initialize our GAN models with weights trained for $\lambda'r + d$ only, which speeds up experiments (compared to training GAN models from scratch) and makes them more controllable. Exact values of all training hyper-parameters are tabulated in Appendix A.6.

We use the non-autoregressive ***Mean&Scale (M&S) Hyperprior*** model from Minnen *et al.* [32] as a strong baseline for an MSE-trained network. We emphasize that this model uses the same probability model $P$ as we use, and that we train it with the same schedule as our models—the main architectural difference to ours is that Minnen *et al.* use a shallower auto-encoder specifically tuned for MSE. We train *M&S Hyperprior* for 15 rate points, allowing us to find a reconstruction with reasonably similar bitrate to our models for all images, and we obtain models outperforming BPG with similar performance to what is reported in [32] (Appendix A.1 shows a rate-distortion plot). When comparing against BPG, we use the highest PSNR setting, *i.e.*, no chroma subsampling and "slow" mode.

**User Study** We compare a total of $N_M$=9 methods: We use *HiFiC* models trained for $r_t \in \{0.14, 0.3, 0.45\}$, denoted ***HiFiC^Lo***, ***HiFiC^Mi***, ***HiFiC^Hi***. For each such model, we go through all images and select the *M&S Hyperprior* model (out of our 15) that produces a reconstruction using *at least* as many bits as *HiFiC* for that image. Additionally, we use *Baseline (no GAN)* trained for $r_t$=0.14, and BPG at two operating points, namely at $1.5\times$ and $2\times$ the bitrate of *HiFiC^Mi*. The resulting bitrates do not exactly match $r_t$ and are shown below models in Fig. 3. We asked $N_P$=14 participants to complete our study. Participants rated an average of 348 pairs of methods, taking them an average of one hour, yielding a total of 4876 comparisons.

## 5 Results

### 5.1 User Study

We visualize the outcome of the user study in Fig. 3. On the x-axis, we show the different methods sorted by the human ranking, with their average bits per pixel (bpp) on the $N_I$ images, as well as the losses used for learned methods. We invert ELO and normalize all scores to fall between 0.01 and 1 for easier visualization. All metrics apart from the inverted ELO are calculated on the entire images instead of user-selected crops as we want to asses the amenability of the metrics for determining ratings and these crops would only be available through user studies.

We can observe the following: Our models (HiFiC) are always preferred to MSE-based models at equal bitrates. Comparing *HiFiC^Lo* to *Baseline (no GAN)*, we can see that adding a GAN clearly helps for human perception. Furthermore, *HiFiC^Lo* at 0.120bpp achieves similar ELO scores as BPG at 0.390bpp ($3.3\times$), and similar scores as *M&S Hyperprior* at 0.405bpp ($3.4\times$). *HiFiC^Mi* at 0.237bpp is preferred to BPG when BPG uses 0.504bpp, more than $2\times$ the bits, and preferred to *M&S Hyperprior* when it uses $1.7\times$ the bits. We note that BPG at around this bitrate is in a regime where the most severe artifacts start to disappear. The fact that *HiFiC^Mi* is preferred to BPG at half the bits shows how using a GAN for neural compression can yield high fidelity images with great bit savings compared to other approaches.

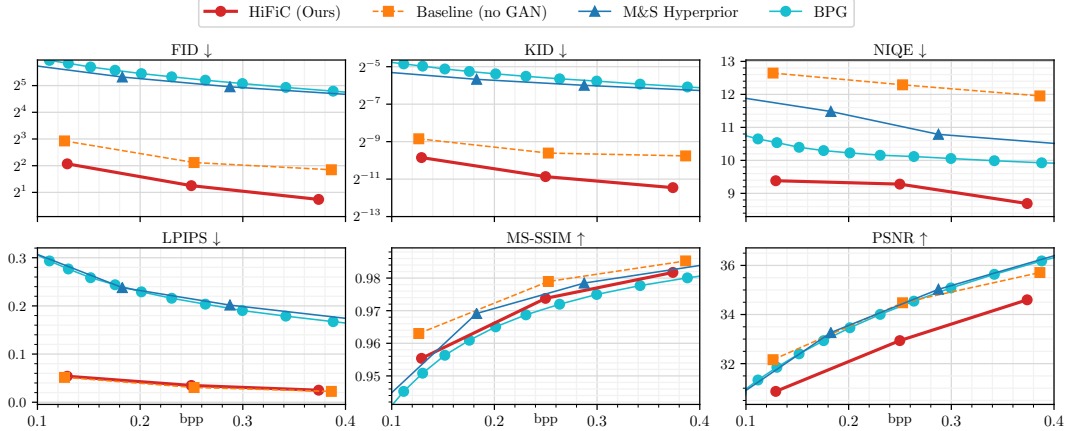

Figure 4: Rate-distortion and -perception curves on CLIC2020. Arrows in the title indicate whether lower is better (↓), or higher is better (↑). Methods are described in Section 4.

Furthermore, if we fix the architecture to ours and the distortion loss to $d = k_M\text{MSE} + k_P d_P$, the perceptual quality indices properly rank the resulting four models (*HiFiC$^{Lo}$*, *HiFiC$^{Mi}$*, *HiFiC$^{Hi}$*, *Baseline (no GAN)*). However, none of the metrics would have predicted the overall human ranking. Especially FID and KID overly punish MSE-based methods, and LPIPS improves when optimized for. On the positive side, this indicates these metrics can be used to order methods of similar architecture and distortion losses. However, we also see that currently there is no metric available to fully replace a user study.

In Appendix A.5, we show that running the Elo tournament per user (inter-participant agreement), and per image (participant consistency at image level) yields the same trends.

## 5.2 Visual Results

**HiFiC**   In Fig. 1, we compare *HiFiC$^{Lo}$* to BPG at the same and at 2× the bitrate. The crops highlight how *HiFiC* reconstructs both the hair and sweater realistically, and very close to the input. BPG at the same bitrate exhibits significant blocking and blurring artifacts, which are lessened but still present at 2× the rate. In the background, we see that our full reconstruction is very similar to the original, including the skin and hat texture. In Appendix B, we show images from all of our datasets and compare to more methods, at various bitrates. There, we also provide download links to all reconstructions. For more visuals, see `hific.github.io`.

**Failure Cases**   In Appendix B, we can also see examples of the two failure cases of *HiFiC*. The first is very small scale text, shown in CLIC2020/25bf4, which looks typeset in another script. The second is faces that are small relative to the image, as in Kodak/14, where our *HiFiC$^{Lo}$* model shows high-frequency noise.

**Previous Work using GANs**   In Appendix A.3, we compare qualitatively to previous work by Agustsson *et al.* [3] and Rippel *et al.* [39].

## 5.3 Quantitative Results

**Effect of GAN**   In Fig. 4, we show rate-distortion and rate-perception curves using our six metrics (see Section 4), on CLIC2020. We compare *HiFiC (Ours)*, *Baseline (no GAN)*, *M&S Hyperprior*, and BPG. We can see that, as expected, our GAN model dominates in all perceptual quality indices, but has comparatively poor PSNR and MS-SSIM. Comparing *Baseline (no GAN)* to *HiFiC*, which both have the same distortion $d = k_M\text{MSE} + k_P d_P$, reveals that the effect of adding a GAN loss is consistent with the theory, and with the user study: all perceptual metrics improve, and both components of $d$ and thus the total distortion $d$ get

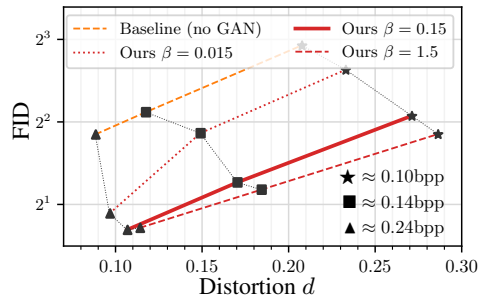

Figure 5: Distortion-perception trade-off.

| Norm in $D$ | Conditional $D$ | FID $\mu$ | $\Delta$ | KID $\mu$ | $\Delta$ | LPIPS $\mu$ | $\Delta$ | NIQE $\mu$ | $\Delta$ |
|---|---|---|---|---|---|---|---|---|---|
| InstanceNorm | | 5.8 | 3.4 | $1.1\mathrm{E}^{-3}$ | $9.7\mathrm{E}^{-4}$ | $6.6\mathrm{E}^{-2}$ | $2.7\mathrm{E}^{-3}$ | 9.0 | 0.66 |
| SpectralNorm | | 4.7 | 0.14 | $1.3\mathrm{E}^{-3}$ | $1.6\mathrm{E}^{-4}$ | $5.6\mathrm{E}^{-2}$ | $1.2\mathrm{E}^{-3}$ | 9.4 | 0.43 |
| InstanceNorm | ✓ | 4.3 | 0.68 | $7.1\mathrm{E}^{-4}$ | $2.4\mathrm{E}^{-4}$ | $6.3\mathrm{E}^{-2}$ | $1.6\mathrm{E}^{-3}$ | 8.8 | 0.64 |
| SpectralNorm | ✓ | 4.3 | 0.36 | $1.2\mathrm{E}^{-3}$ | $2.2\mathrm{E}^{-4}$ | $5.5\mathrm{E}^{-2}$ | $7.4\mathrm{E}^{-4}$ | 9.4 | 0.32 |

Table 1: Exploring across-run variation. Each model (row) is run four times in exactly the same configuration, and we show mean $\mu$ and difference between maximal and minimal value $\Delta$.

worse (see also next paragraph). We observe that MSE (PSNR) is more at odds with the GAN loss than LPIPS, which gets only marginally worse when we add it. These observations motivate us to use FID in ablation studies, as long as $d$ and the overall training setup is fixed. We show similar figures for the other datasets in Appendix A.8.

**Distortion-Perception Trade-off** In Fig. 5, we show how varying the GAN loss weight $\beta$ navigates the distortion-perception trade-off. We plot FID on the y axis, and the full $d = k_M\mathrm{MSE} + k_P d_P$ on the x axis. We show *HiFiC* on a exponential grid $\beta \in \{0.015, 0.15, 1.5\}$, and *Baseline (no GAN)* as a reference. Each model is shown at three different bitrates, and models with similar bitrate are connected with black dotted lines. We see that across rates, as we increase $\beta$, the perceptual quality index improves, while $d$ suffers. This effect is lessened at higher rates (towards left of figure), possibly due to different relative loss magnitudes at this rate, *i.e.*, $d$ is much smaller, $r$ is larger.

## 5.4 Studies

**Discriminator: On Conditioning and Normalization** Recall that we use a conditional $D$, in contrast to previous work. While our formulation can be adapted to include a non-conditional $D$, this changes training dynamics significantly: Theoretically, ignoring the distortion $d$ for a moment, $G$ can then learn to produce a natural image that is completely unrelated do with the input, and still achieve a good discriminator loss. While we guide $G$ with $d$, we nevertheless found that training a *non-conditional* discriminator leads to images that are less sharp. The resulting models also obtain a worse FID, see Fig. 6a.

In [51], InstanceNorm is used in $D$, which causes artifacts in our setup, prompting us to replace it with SpectralNorm [36]. In pure generation, SpectralNorm was shown to reduce the variance of FID across multiple runs [24]. To explore this in our setup, we run each of the models shown in Table 1 four times, and measure the mean and largest difference between runs, on our metrics. We find that in the *non-conditional* setting, SpectralNorm reduces the variability of FID and KID significantly, and, to a lesser extent, also that of LPIPS and NIQE. This effect is weakened when using the *conditional* $D$, where SpectralNorm only marginally reduces variability in all metrics, which we view as a further benefit of conditioning. Overall, keeping the normalization dimension fixed, we also see that conditioning $D$ improves all metrics.

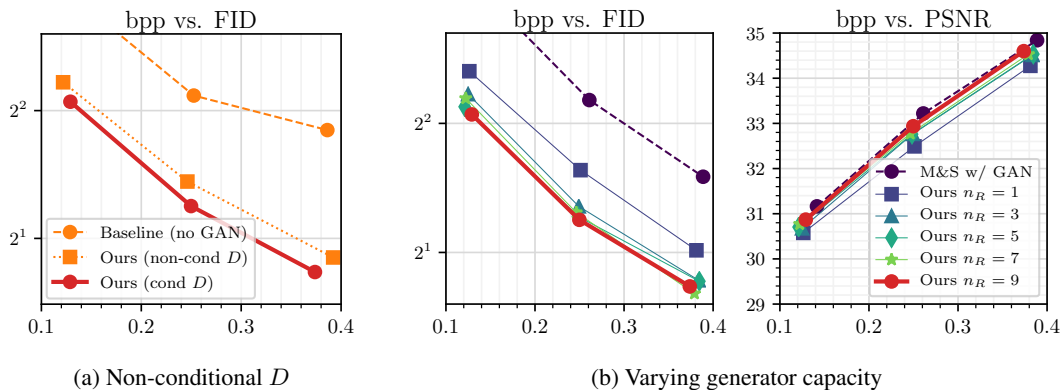

(a) Non-conditional $D$        (b) Varying generator capacity

Figure 6: a) Shows the effect of a non-conditional $D$, b) shows models with different number of residual blocks $n_R$ in $G$.

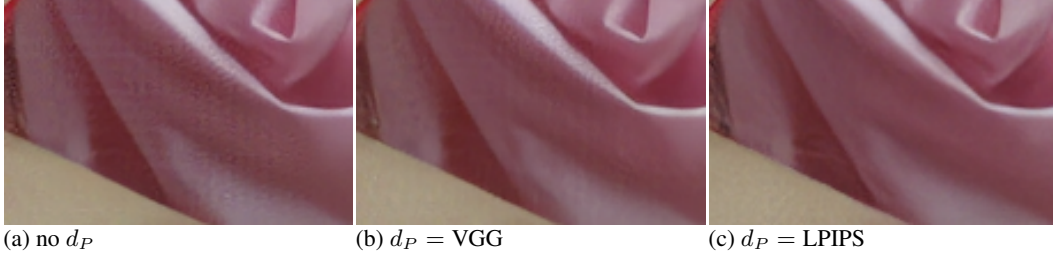

(a) no $d_P$            (b) $d_P$ = VGG           (c) $d_P$ = LPIPS

Figure 8: Effect of varying the perceptual distortion $d_P$. All models were also trained with an MSE loss and a GAN loss.

**Instance Norm in Auto-Encoder** We visualize the darkening caused by InstanceNorm in $E, G$ (see Section 3.3) in the inset figure, where a model is evaluated on an image at a resolution of $512{\times}512$px as well as at the training resolution ($256{\times}256$px). We also explored using BatchNorm [19] but found that this resulted in unstable training in our setup.

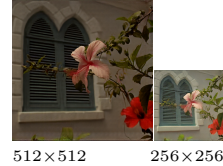

$512{\times}512$      $256{\times}256$

**Generator Capacity** By default, we use $n_R$=9 residual blocks in $G$ (see Fig. 2). We show the result of varying $n_R$ in Fig. 6b. While $n_R$=1 gives significantly worse FID and PSNR, both metrics start to saturate around $n_R$=5. We also show the *M&S Hyperprior* baseline trained with our loss ("M&S w/ GAN"), *i.e.*, with LPIPS and a conditional $D$, and exactly the same training schedule. While this yields slightly better PSNR as HiFiC, FID drops by a factor 2, further indicating the importance of capacity, but also how suited the hyperprior architecture is for MSE training.

**Training Stability and Losses** Training GANs for unconditional or class-conditional image generation is notoriously hard [29, 36, 10]. However, our setup is inherently different through the distortion loss $d$, which guides the optimization at the pixel-level. While our initialization scheme of using weights trained to minimize $\lambda' r + d$ (see Section 4) can help, MSE seems to be the driving force for stability in our setup: For Fig. 9, we initialize the network with $\lambda' r + d$-optimized weights, then train with a GAN loss but without a distortion loss $d$. We compare fixing $E$ to using the $\lambda' r + d$ weights, to jointly learning $E$. We see that the GAN collapses if we learn $E$. Additionally, we explore varying $d_P$: In Fig. 8a, we see that removing $d_P$ introduces significant gridding artifacts. Using $d_P$=VGG, *i.e.*,

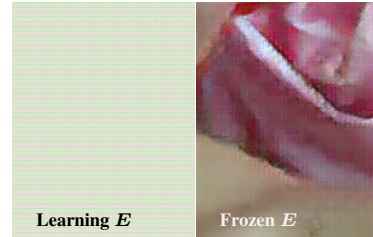

**Learning $E$**      Frozen $E$

Figure 9: Training with a GAN loss, but no MSE or $d_P$.

$L_1$ in the feature space of VGG, some gridding is still visible (Fig. 8b). Using $d_P$=LPIPS alleviates these artifacts. We note that only using LPIPS *without a GAN* also produces minor repetitive patterns as well as less sharp reconstructions, which is validated by the fact that *Baseline (no GAN)* ranks worse than our GAN models in the user study. We show a visual example in Appendix A.2.

## 6 Conclusion

In this paper, we showed how optimizing a neural compression scheme with a GAN yields reconstructions with high perceptual fidelity that are visually close to the input, and that are preferred to previous methods even when these approaches use more than double the bits. We evaluated our approach with a diverse set of metrics and interpreted the results with rate-distortion-perception theory. Guided by the metrics, we studied various components of our architecture and loss. By comparing our metrics with the outcome of the user study, we showed that no existing metric is perfect for ordering arbitrary models right now, but that using FID and KID can be a valuable tool in exploring architectures and other design choices. Future work could focus on further studying perceptual score indices and metrics (to better predict human preferences), and further investigate failure cases such as small faces. Additionally, generative video compression is a very interesting direction for future work. To ensure temporal consistency, mechanisms to enforce smoothness could be borrowed from, e.g., GAN-based video super resolution models.

## Broader Impact

Users of our compression method can benefit from better reconstructions at lower bitrates, reducing the amount of storage needed to save pictures and the bandwidth required to transmit pictures. The latter is important as wireless technology typically lags behind user trends which have been continuously requiring higher bandwidths over the past decades, and there is no end in sight with emerging applications such as virtual reality. Furthermore, better compression technology improves accessibility in developing areas of the world where the wireless infrastructure is less performant and robust than in developed countries. It is important to keep in mind that we employ a generator $G$ that in theory can produce images that are very different from the input. While this is the case for *any* lossy image compression algorithm, this has a bigger impact here as we specifically train $G$ for realistic reconstructions. Therefore, we emphasize that our method is not suitable for sensitive image contents, such as, *e.g.*, storing medical images, or important documents.

## Funding

This work was done at Google Research.

## Acknowledgments

The authors would like to thank Johannes Balle, Sergi Caelles, Sander Dielmann, and David Minnen for the insightful discussions and feedback.

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
