[Supplementary Material]

# A  Supplementary Material – High-Fidelity Generative Image Compression

We show further details of our method in A.1–A.9, and more visual results in Appendix B.

## A.1  Comparing MSE models based on Minnen *et al.* [32]

Figure A1: Comparing the *Mean & Scale Hyperprior* model trained by us to the one reported by Minnen *et al.* [32], and to their main model, denoted "Minnen *et al.* 2018".

To validate that the *M&S Hyperprior* model trained by us is a strong MSE model, we compare it to models reported by Minnen *et al.* [32] in Fig. A1. First, we compare the model trained by us to the "Mean & Scale Hyperprior" baseline reported in [32], which is equivalent to the *M&S Hyperprior* model in terms of architecture. We observe a very minor drop in PSNR, which is likely attributable to our training schedule of $2\,000\,000$ steps, vs. the $6\,000\,000$ steps used in [32]. Then, we compare to the fully autoregressive main model of [32], denoted "Minnen *et al.* 2018". We can see that this model is on average $\approx 0.4$dB better than our *M&S Hyperprior* model. It is important to note that a) an auto-regressive probability model would also increase the performance of all of our models, and b) that in the user study, our GAN models were preferred to MSE-trained Hyperprior models even when the Hyperprior models used $4\times$ the bitrate, which amounts roughly to a $\approx 4$dB PSNR difference.

## A.2  Using LPIPS without a GAN loss

We saw in the user study that the model trained for MSE and LPIPS without a GAN loss (*Baseline (no GAN)*) ranks worse than the one using a GAN loss (*HiFiC$^{Lo}$*). In Fig. A2, we visualize a reconstruction of *Baseline (no GAN)* to show that this training setup can also cause gridding artifacts on some images. Furthermore, we see that adding a GAN loss leads to sharper reconstructions and to a faithful preservation of the image noise.

Orignal       $r$, MSE, LPIPS, GAN (*HiFiC$^{Lo}$*)       $r$, MSE, LPIPS (*Baseline (no GAN)*)

Figure A2: LPIPS without a GAN (right) leads to gridding and less sharpness than using a GAN (middle). We show the loss components below the figure, where $r$ is the rate loss. *Best viewed on screen.*

## A.3    Qualitative Comparison to Previous Generative Approaches

We compare against Agustsson *et al.* [3] as well as Rippel *et al.* [39] in Fig. A3 on an image from the Kodak dataset, as both incorporated adverserial losses into their training. We do no further comparisons, as Agustsson *et al.* targeted "extremely low" bitrates and thus operate in a very different regime than our models, and Rippel *et al.* only released a handful of images.

Original                        HiFiC$^{Lo}$ (Ours): 0.162bpp

Rippel *et al.* [39]: 0.194bpp          Agustsson *et al.* [3]: 0.0668bpp

Figure A3: Comparison between neural compression approaches using an adverserial loss on Kodak/19. Note the high-frequency artifacts in the tower and fence for [39], as well as the different grass texture. Note that the comparison to [3] is not ideal, as their highest bpp reconstructions use less than half the rate of *HiFiC*, but we see that the reconstruction deviates significantly from the input. Furthermore, both [39] and [3] exhibit a small color shift.

Figure A4: Visualizing which axes different normalization layers normalize over. $N$ is the batch dimension, $C$ the channel dimension, and $H, W$ is the spatial dimensions. If the shaded area spans over an axis, this axis is normalized over. For example, BatchNorm normalizes over space and batches, LayerNorm over space and channels. Our normalization layer, ChannelNorm, normalizes over channels only. *Figure adapted from [55].*

## A.4 ChannelNorm

Fig. A4 shows a visual comparison of different normalization layers, which is based on the visualization provided in [55]. We note that BatchNorm [19], LayerNorm [4], InstanceNorm [49], and GroupNorm [55], all average over space.

We compare ChannelNorm to InstanceNorm via the following equations. To simplify notation, we assume that normal broadcasting rules apply, *e.g.*, $f_{chw} - \mu_c$ means that when calculating channel $c$, we subtract $\mu_c$ from each spatial location $h, w$ in the $c$-th channel of $f$. As introduced in Section 3.3, ChannelNorm uses learned per-channel offsets $\alpha_c, \beta_c$ and normalizes over channels:

$$f'_{chw} = \frac{f_{chw} - \mu_{hw}}{\sigma_{hw}}\alpha_c + \beta_c, \qquad \mu_{hw} = {}^1\!/\!c \sum_{c=1}^{C} f_{chw}$$

$$\sigma_{hw}^2 = {}^1\!/\!c \sum_{c=1}^{C} (f_{chw} - \mu_{hw})^2.$$

InstanceNorm also uses $\alpha_c, \beta_c$, but normalizes spatially:

$$f'_{chw} = \frac{f_{chw} - \mu_c}{\sigma_c}\alpha_c + \beta_c, \qquad \mu_c = {}^1\!/\!HW \sum_{h=1}^{H} \sum_{w=1}^{W} f_{chw}$$

$$\sigma_c^2 = {}^1\!/\!HW \sum_{h=1}^{H} \sum_{w=1}^{W} (f_{chw} - \mu_c)^2.$$

We hypothesize that the dependency on $H, W$ causes the generalization problems we see.

## A.5  User Study: More Results

For the user study plot in the main text (Fig. 3), we averaged Elo scores across all raters, across all images, *i.e.*, the Elo tournament is over all comparisons (see Section 3.4). This data is visualized as a box plot in Fig. A5.

In this section, we also show the result of averaging over participants in Fig. A6 (running an Elo tournament per participant), and over images in Fig. A7 (running an Elo tournament per image). As we can see, the overall order remains unchanged, except for the three methods that were very close in median Elo score in Fig. 3 (HiFiC at 0.120bpp, BPG at 0.390bpp, and M&S Hyperprior at 0.405bpp), which change their order. Also, different images obtain a wide range of Elo scores.

In Table 2, we show the first 5 characters of the names of the $N_I = 20$ images used for the user study, and the rank each method scored on this image, where lower is better. We note that for all images, one of our GAN models earns first or second place – except for the last image. This image contains a lot of small scale text, and is shown as part of our visualizations in Appendix B. There, we also provide a link to all images used in the user study.

In Fig. A9, we show a screenshot of the GUI shown to raters. The crops selected by the participants are available at `https://hific.github.io/raw/userstudy.json`.

Figure A5: Global Monte Carlo Elo Scores. We use the standard box plot visualization: The horizontal green thick line indicates the median, the box extends from Q1 to Q3 quartiles, the whiskers extend to $1.5 \cdot (Q3 - Q1)$, and outlier points are points past the whiskers.

Figure A6: Per-participant Monte Carlo Elo Scores. We use the standard box plot visualization: The horizontal green thick line indicates the median, the box extends from Q1 to Q3 quartiles, the whiskers extend to $1.5 \cdot (Q3 - Q1)$, and outlier points are points past the whiskers.

**Figure A7:** Per-image Monte Carlo Elo Scores. We use the standard box plot visualization: The horizontal green thick line indicates the median, the box extends from Q1 to Q3 quartiles, the whiskers extend to $1.5 \cdot (Q3 - Q1)$, and outlier points are points past the whiskers.

| Image name | HiFiC$^{Hi}$ 0.359 bpp | HiFiC$^{Mi}$ 0.237 bpp | BPG 0.504 bpp | M&S 0.405 bpp | HiFiC$^{Lo}$ 0.120 bpp | BPG 0.390 bpp | M&S 0.272 bpp | no GAN 0.118 bpp | M&S 0.133 bpp |
|---|---|---|---|---|---|---|---|---|---|
| e0256 | 1 | 2 | 3 | 5 | 4 | 6 | 8 | 7 | 9 |
| a251f | 1 | 2 | 4 | 8 | 3 | 5 | 7 | 6 | 9 |
| 0ae78 | 1 | 2 | 3 | 6 | 5 | 4 | 8 | 7 | 9 |
| 95e7d | 1 | 2 | 3 | 6 | 4 | 5 | 8 | 7 | 9 |
| 2145f | 1 | 2 | 4 | 3 | 5 | 6 | 7 | 8 | 9 |
| 58c13 | 2 | 1 | 3 | 5 | 6 | 4 | 7 | 9 | 8 |
| f063e | 2 | 4 | 1 | 6 | 5 | 3 | 7 | 8 | 9 |
| dcb53 | 1 | 2 | 4 | 3 | 5 | 6 | 7 | 8 | 9 |
| d5424 | 1 | 2 | 3 | 5 | 6 | 4 | 8 | 7 | 9 |
| 72e19 | 1 | 2 | 5 | 4 | 3 | 8 | 6 | 7 | 9 |
| 1c55a | 1 | 5 | 2 | 4 | 6 | 3 | 7 | 8 | 9 |
| ad249 | 2 | 1 | 5 | 4 | 3 | 7 | 8 | 6 | 9 |
| d9692 | 2 | 5 | 1 | 4 | 6 | 3 | 7 | 8 | 9 |
| 18089 | 1 | 3 | 2 | 5 | 6 | 4 | 7 | 8 | 9 |
| f7a9e | 1 | 2 | 4 | 3 | 5 | 6 | 7 | 8 | 9 |
| a09ce | 1 | 3 | 2 | 5 | 7 | 4 | 6 | 8 | 9 |
| 6e8e3 | 1 | 2 | 4 | 5 | 3 | 6 | 8 | 7 | 9 |
| afa0a | 1 | 2 | 4 | 6 | 3 | 5 | 7 | 8 | 9 |
| 8ba19 | 2 | 1 | 3 | 5 | 4 | 6 | 7 | 8 | 9 |
| 25bf4 | 4 | 6 | 1 | 3 | 8 | 2 | 5 | 7 | 9 |

**Table 2:** Per image rankings of the user study. We show the average bpp of each method in the header

## A.6 Training Details

We follow standard practice of alternating between training $E$, $G$, $P$ (jointly) for one step and training $D$ for one step. We use the same learning rate of $1\text{E}{-4}$ for all networks, and train using the Adam optimizer [22]. At the beginning of training, the rate loss can dominate. To alleviate this, we always first train with a higher $\lambda$, as in [32]. As mentioned in Section 4, we initialize our GAN models (HiFiC$^{Hi}$, HiFiC$^{Mi}$, HiFiC$^{Lo}$) from a model trained for MSE and $d_P$ =LPIPS. Table 3 shows our different models and the LR and $\lambda$ schedules we use. The GAN initialization is using the *Warmup* model. Together, this yields 2M steps for all models.

We fix hyper-parameters shown in Fig. A8a for all experiments (unless noted), and vary $\lambda^{(a)}$ depending on $r_t$ as shown in Fig. A8b.

The training code and configs for *HiFiC$^{Lo}$* and *Baseline (no GAN)* is available at `hific.github.io`.

| | Losses | Initialize with | Training | LR decay | Higher $\lambda$ |
|---|---|---|---|---|---|
| Baseline (no GAN) | MSE+LPIPS | - | 2M steps | 1.6M steps | 1M steps |
| M&S Hyperprior | MSE | - | 2M steps | 1.6M steps | 1M steps |
| *Warmup* | MSE+LPIPS | - | 1M steps | 0.5M steps | 50k steps |
| HiFiC [Hi,Mi,Lo] | MSE+LPIPS+GAN | *Warmup* | 1M steps | 0.5M steps | 50k steps |

Table 3: Training schedules, using "M" for million, "k" for thousand.

| Parameter | Value | Note |
|---|---|---|
| $\lambda^{(b)}$ | $2^{-4}$ | |
| $k_M$ | $0.075 \cdot 2^{-5}$ | |
| $k_P$ | 1 | |
| $C_y$ | 220 | |
| $\beta$ | 0.15 | Except for Fig. 5 |

| Target Rate $r_t$ | $\lambda^{(a)}$ |
|---|---|
| 0.14 | $2^1$ |
| 0.30 | $2^0$ |
| 0.45 | $2^{-1}$ |

(a) Fixed hyper-parameters.      (b) Varying hyper-parameters.

Figure A8: Hyper-parameters.

## A.7 Patch-based FID and KID

As mentioned in Section 4, we extract patches to calculate FID and KID. From each $H \times W$ image, we first extract $\lfloor H/f \rfloor \cdot \lfloor W/f \rfloor$ non-overlapping $f \times f$ crops, and then shift the extraction origin by $f/2$ in both dimensions to extract another $(\lfloor H/f \rfloor - 1) \cdot (\lfloor W/f \rfloor - 1)$ patches. We use $f = 256$ in all evaluations.

## A.8 Further Quantitative Results

In this section, we provide plots similar to Fig. 4 on the other two datasets: In Fig. A10, we show rate-distortion and rate-perception curves for DIV2K [2], and in Fig. A11, we show curves for Kodak [23]. As noted in Section 4, the 24 Kodak images only yield 192 patches to calculate FID and KID, and we thus omit the two metrics.

## A.9 Image Dimensions of the Datasets

We use the three datasets mentioned in Section 4 for our evaluation. Kodak contains images of $768 \times 512$ pixels, the other two datasets contain images of varying dimensions. To visualize the distribution of these dimensions, we show histograms for the shorter dimensions, for the product of both dimensions, and for the aspect ratio in Fig. A12. We see that most images cluster around shorter sides of 1400px, and go up to 2000px. We note that the three biggest images for CLIC are of dimensions $2048 \times 2048, 2048 \times 2048, 2000 \times 2000$, for DIV2K they are $2040 \times 2040, 1872 \times 2040, 1740 \times 2040$.

Figure A9: Screenshot of the user study GUI.

Figure A10: Rate-distortion and -perception curves on DIV2K. Arrows in the title indicate whether lower is better (↓), or higher is better (↑).

Figure A11: Rate-distortion and -perception curves on Kodak. Arrows in the title indicate whether lower is better (↓), or higher is better (↑).

Figure A12: Histograms to show image dimensions for CLIC2020 and DIV2K.

# B   Further Visual Results

## B.1   PDF Visualization

*Due to size constraints, the PDF is hosted at* `https://hific.github.io/appendixb`.

In the PDF, we show images from all datasets. For each image, we show the full reconstruction by one of our models, next to the original. On the top left of this image, we show the dataset and image ID. Additionally, we pick one or two crops for each image, where we compare the original to the following methods:

1. HiFiC$^{\text{Mi}}$ (Ours)
2. HiFiC$^{\text{Lo}}$ (Ours)
3. *M&S Hyperprior* at a bitrate greater than *HiFiC$^{Lo}$*
4. BPG at the $2\times$ the bitrate of *HiFiC$^{Mi}$*
5. BPG at the same bitrate as *HiFiC$^{Mi}$*
6. BPG at the same bitrate as *HiFiC$^{Lo}$*
7. JPEG at $Q = 80$.

We chose to add JPEG at $Q = 80$ as a further reference, as this is a quality factor in common use [13].

**Results**

Throughout the examples, we can see that our GAN models shine at reconstructing plausible textures, yielding reconstructions that are very close to the original. We see that BPG at the same bitrate as *HiFiC$^{Lo}$ (Ours)* tends to exibit block artifacts, while *HiFiC$^{Lo}$* looks very realistic. For most images, our *HiFiC$^{Mi}$* model also looks significantly better than BPG at the same bitrate. When BPG uses $2\times$ the rate as *HiFiC$^{Mi}$*, it starts to look similar to our reconstructions.

## B.2   More Comparisons and Raw Images

We note that the crops are embedded as PNGs in the PDF, but the large background images are embedded as JPEGs to prevent a huge file. However, we package full-sized PNGs for various methods into ZIPs. The ZIP files also contain a HTML file that can be used for easy side-by-side comparisons, as this is the best way to visualize differences.

The images are also available directly in the browser at:

`https://hific.github.io/raw/index.html`.

The ZIP files:

1. The following ZIP contains the 20 images from CLIC2020 used for the user study, compressed with each of the 9 methods used in the study.
   `https://hific.github.io/raw/userstudy`
   Size: 610MB

2. The following ZIP files contain all the images of the respective datasets, compressed with *HiFiC$^{Hi}$*, *HiFiC$^{Mi}$*, *HiFiC$^{Lo}$*.
   `https://hific.github.io/raw/kodak`
   Size: 61MB
   `https://hific.github.io/raw/clic2020`
   Size: 6.3GB
   `https://hific.github.io/raw/div2k`
   Size: 1.7GB