[Reviews · NeurIPS 2020]

Review 1

Summary and Contributions: The authors study how different elements of an optimization algorithm effect optimized generative models for lossy image compression (including architectural decisions, training strategies, and perceptual loss functions). They claim their results achieve a state of the art (evaluated with both perceptual metrics and a user study) and also say that their methodology works on high resolution images.

Strengths: --- Updated Review --- I want to thank the authors for addressing some of my concerns. I believe as I did before that this is quite a strong submission and my review of it is unchanged. A clear accept. This is a very strong submission with both extremely compelling practical results, and interesting theoretical progress and observations that advance our understanding of why previous methods have had the types of failures that they have had. I believe this is of high significance, novelty and relevance to the entire NeurIPS community, especially those working in image compression and perceptual metrics and loss functions.

Weaknesses: I'm a bit concerned by the perceptual study methodology. The authors ask users to search for "interesting" areas of the image that will be shown Is it assumed that there are no meaningful compression errors in flat/featureless areas of the image? Artifacts in these areas could be the most noticeable artifacts from a pure perception standpoint (as they are not masked by local contrast though may be masked by luminance). Additionally, are any considerations taken into account to ensure consistency of display and viewing conditions (ie resolution, color space, size of display, viewing distance, lighting conditions? ) It's also a bit strange to compare full image ratings from the metrics to crops rated by the individuals. Can the authors address this?

Correctness: I'd like the authors to address the concerns listed above with the user study, but I believe the visible results to be quite impressive regardless of the quantification.

Clarity: Please fix the labeling on figure 3 (cannot see some of the text below the x axis). The axes in figure 5 are not labeled and thus the figures are not entirely clear (though one can guess what they are). Please fix and make more clear.

Relation to Prior Work: Prior work is clearly discussed, critiqued, built upon and it is shown how this work moves to a novel contribution.

Reproducibility: Yes

Additional Feedback:


Review 2

Summary and Contributions: The authors present a method for lossy neural image compression that combines conditional generative adversarial networks with learned compression. The method is able to achieve competitive rate distortion performance perceptually as evidenced by an extensive evaluation. It also operates at fairly high resolutions (e.g. 2k x 2k) and the authors additionally show an analysis of the different architectural components.

Strengths: The topic of neural image compression is currently very significant and gains in rate distortion performance through new methods would have a positive impact e.g. on bandwidth used regarding transmission of visual content. In my opinion, the strength of the paper is to propose a system that achieves good practical performance when measuring rate distortion performance perceptually. The evaluation seems thorough and the results look very convincing outperforming previous compression methods that employ generative adversarial networks. The analysis of the individual components used is sufficiently detailed.

Weaknesses: While the approach seems sound and practically achieves good performance, the idea of combining generative adversarial networks and learned compression is (also as the authors show in related work) not new. The novelty here lies in the details of how the system is designed exactly and arguably in the conditioning of the GAN. Despite a potential lack in novelty, I still tend towards acceptance since the system is setting a new state of the art and reading the paper is insightful.

Correctness: The claims and method look correct to me.

Clarity: The paper is well written.

Relation to Prior Work: Related work is clearly discussed.

Reproducibility: Yes

Additional Feedback: With VVC being finalised, it would be interesting to know how this neural image compression method compares to IFrame coding in VVC. If this is still too early to do such comparisons, I do not insist to include this. To get a better feeling of the reconstruction the GAN encourages, I would be interested to see what happens temporally when compressing an image sequence. Do the authors expect temporal artefacts or do smooth changes of the input carry over to temporally coherent output? In case there are temporal instabilities, are they only observed for a certain low bit rate?


Review 3

Summary and Contributions: The submission presents an end-to-end learned lossy image compression model that achieves state of the art bit-rates while maintaining high perceptual quality. Update after rebuttal: I am happy with the promised inclusion and discussion of failure modes in the main text and maintain my positive assessment.

Strengths: - Impressive visual quality of compressed images - Extensive and sound evaluation showing clear average performance improvements over classical image compression methods as well end-to-end learned methods.

Weaknesses: - Little conceptual novelty, mainly skillful engineering using existing model components. - No examples and discussion of model limitations in the main text. As typical for GAN-based image synthesis, the model has issues generating fine-detail that is not texture. This becomes particularly visible for human bodies/faces (Appendix page 23, Kodak/14) or text (Appendix page 31, CLIC 25bf4). In both cases I would argue that the comparable BPG compression is clearly superior. I believe that these quality shortcomings for important sub-categories of images are a major obstacle to deploying such models as a generic image compression algorithm in real world settings and thus should be adequately discussed in the main text.

Correctness: Yes

Clarity: Yes

Relation to Prior Work: Yes

Reproducibility: Yes

Additional Feedback: n/a


Review 4

Summary and Contributions: This paper proposes a generative compression method to achieve high quality reconstructions, In a user study, the paper shows that the proposed approach is visually preferred. Several distortion metrics are used to evaluate the method quantitatively. and it extensively studies the proposed architecture, training strategies, as well as the loss, in terms of perceptual metrics and stability.

Strengths: This paper improves the GAN based image compression method using the conditional GAN, appling a perceptual distortion and modifies several layers. User study experiments and evaluation using different metrics are clarified specifically.

Weaknesses: Two comments are listed as below: 1) As the title named High-Fidelity Generative Image Compression, I can not understand why it is called high-fidelity. With the GAN loss and perceptual loss added, the compressed image can be more perceptually preferred as proved before. However, texture generated by GAN is usually fake and it may not be called high-fidelity. I think an additional section is needed to discuss texture generated by the proposed method is much closer to input than others apart from perceptual evaluation. 2) Comparison with the prior work which utilized GAN is necessary to include in section 4 to illustrate the advantage of the proposed method.

Correctness: Yes.

Clarity: Yes.

Relation to Prior Work: The proposed one utilizes Conditional GAN instead of GAN, single scale instead of multi-scale, channel-averaging LayerNorm instead of InstanceNorm and additionally a perceptual distortion. However, it may lack of proof of the above changes. Maybe ablation study is necessary.

Reproducibility: Yes

Additional Feedback:

[Author Response · NeurIPS 2020]

# High-Fidelity Generative Image Compression – Rebuttal

We thank the reviewers for the positive feedback. In the paper, we extensively study how to combine Generative Adversarial Networks and learned compression to obtain a state-of-the-art generative lossy compression system.

As the reviewers state, our method yields "impressive visual quality" (R3), "high quality reconstructions" (R4), the paper contains "extensive and sound evaluation" (R3) and "extensively studies the proposed architecture, training strategies, as well as the loss" (R4). We show "extremely compelling practical results, and interesting theoretical progress and observations" (R1). The paper is "well written" (R2) and "insightful" (R2). Additionally, "prior work is clearly discussed, critiqued, built upon and it is shown how this work moves to a novel contribution" (R1).

In the following, we address the questions and concerns raised by the reviewers in detail.

**R1: Concerns about user study methodology.** *"The authors ask users to search for 'interesting' areas of the image [...]. Is it assumed that there are no meaningful compression errors in flat/featureless areas of the image?"* We chose crop size such that the crops are large enough to contain context, while at the same time focusing raters on a part of the image (rating full-resolution images is much harder, as the rater's attention is divided between parts of the image). This results in relatively large (768×768px crops – see Appendix A.8 for a visual example of the study interface) which also contain flat areas. Furthermore, we observe our method actually shines in flat regions (see, e.g., the wall in image CLIC2020/b3f37 in Appendix B). We will release the crop location selected by each user, for each image, for other researchers to compare.

*"It's also a bit strange to compare full image ratings from the metrics to crops rated by the individuals."* We want to assess whether any metric gets rid of the need to do a user study (L216). When using a metric without a study, user-selected crops would not be available. As to why we use crops and not the full image for raters: it allows us i) to focus the rater's attention (see previous reply) ii) to avoid any downsampling (which would bias results) iii) to show the original next to the reconstruction. Note that each rater selects a different random crop, which together cover most of the image.

*"Do you ensure consistency of display and viewing conditions?"* While we did not ensure consistency of viewing distance and color space, the study participants were all professionals with access to a well lit office and a modern, sufficiently large, and bright monitor.

**R4: Comparison with prior work needed in main text.** As we note in L29, Agustsson et al. targeted "extremely low" bitrates and thus operate in a different regime than our models. We nevertheless showed a qualitative comparison in Appendix A.3. There, we also compare to Rippel et al., who did not release models or sufficient images to calculate statistics. We will add a pointer to that Appendix in the main text.

**R2: If possible, it would be interesting to compare to VVC.** VVC licensing prohibits us from running that code (if we read the licensing terms correctly). We are open to adding a comparison if someone with the proper access rights could run VVC for us (acknowledging them, of course). Since we will open source the model, and the reconstructions, this should be very easy to do for such a person/entity.

**R2: What happens when you apply HiFiC to an image sequence?** This is an interesting question. Because we focused on images, our method contains no mechanism to guarantee temporal consistency, and inconsistencies in fine detail between frames will likely be visible. In general, we believe that generative video compression is a very interesting direction for future work, and to ensure temporal consistency, mechanisms to enforce smoothness could be borrowed from, e.g., GAN-based video super resolution models. We will mention this in the conclusion.

**R3: Discuss failure cases in main text.** We will include the relevant part of Appendix B where we discuss the few failure cases ("very small scale text [and] small scale faces" L534–L536) in the main text, and extend it, as part of the additional 9th content page available for the camera ready version. We are interested in seeing how future work addresses these limitations, and will highlight this direction in the conclusion. One could, e.g., include a face detector in the training pipeline, and raise the MSE penalty in the face region, or fall back to other approaches in these regions. In this paper, we focused on improving overall visual quality across a wide variety of images while keeping the method relatively simple.

**R4: Why "high-fidelity"?** The reviewer is concerned that GANs produce "fake" texture, and finds that the title may thus not be appropriate. We agree that the proposed method may not be pixel-wise accurate to the input, however, the reconstructions are very close semantically and texture-wise. This is validated by the user study (users see the original), as well as the visuals in the main text and Appendix B. We thus think it is justified to call our approach "high-fidelity" in the sense of matching the input image distribution. We also note that "high-fidelity" is commonly used with this meaning in the GAN literature, e.g., in Bock, Donahue, and Simonayan's "Large Scale GAN Training for High Fidelity Natural Image Synthesis" (ICLR 2019, arxiv 1809.11096). We will clarify this perspective in the paper. If the reviewers insist, we could of course change the title, to, e.g., "High-Resolution Generative Image Compression".

[Meta-Review · NeurIPS 2020]

The paper has been very well received by the reviewers for its novelty and the strong experimental evaluation. The ability to handle large scale inputs and the high compression ratios reported make this work a great contribution for the community. I would encourage the authors to make the final effort in further improving their work by addressing remaining concerns of the reviewers, one being discussion on limitations of the approach for instance. Eventual further comparisons with GAN based approaches would also be desired but not a required change.